# Analysis of the Modulation of Dormancy Release in Almond (*Prunus dulcis*) in Relation to the Flowering and Ripening Dates and Production under Controlled Temperature Conditions

**Ángela S. Prudencio, Pedro Martínez-Gómez *** and **Federico Dicenta**

Department of Plant Breeding, CEBAS-CSIC, PO Box 164, 30100 Espinardo, Murcia, Spain;
asanchez@cebas.csic.es (Á.S.P.); fdicenta@cebas.csic.es (F.D.)

**\*** Correspondence: pmartinez@cebas.csic.es; Tel.: +34-968-396-200

**Abstract:** In this study, the control of eco- and endo-dormancy release led to the modulation of the flowering time in almond (*Prunus dulcis* (Mill.) D.A. Webb). The study was performed in almond cultivars with contrasting flowering times: the extra-early flowering cultivar Desmayo Largueta and the ultra-late cultivar Tardona. Temperature control in the "Autumn", "Winter" and "Spring" chambers successfully delayed the flowering time in Desmayo Largueta. Advance flowering in the cultivar Tardona was limited, however, even with the application of sufficient chill in the Winter chamber. The ecodormancy period and the heat accumulation for flowering were not stable among cultivars, even though the heat accumulation was generally high, in accordance with that accumulated in field conditions. The heat requirements of the early cultivar Desmayo Largueta were lower than those of the ultra-late cultivar Tardona. We observed a decreasing pattern in ecodormancy along treatments that was probably related to the temperature in the Spring chamber. Finally, flowering and fruit set were highly variable, and these parameters were more dependent on the cultivar assayed than on the treatment applied. Although the ripening time under our experimental conditions was earlier than the phenological dates observed in the field, the flowering time delayed the ripening time in the case of the extra-early cultivar Desmayo Largueta. The fruit weight increased in the last treatments, whereas the kernel/fruit ratio decreased, as the kernel weight did not vary significantly along treatments. The results obtained show that flowering time can be modulated by temperature control and that other uncontrolled factors, such as photoperiod, can be involved in the control of endodormancy release and flowering time, especially in late flowering cultivars.

**Keywords:** almond; *Prunus*; flowering; dormancy; production; breeding; climate change

## 1. Introduction

Dormancy in fruit tree species is described as a rest period determined firstly by endogenous factors (the endodormancy phase) and later by exogenous factors (the ecodormancy phase). The length of the endodormancy period is dependent on the chilling requirements of each cultivar, whereas the ecodormancy period is dependent on the heat requirements to flower. Both endodormancy and ecodormancy determine the flowering time [1]. Late flowering advances in almond imply a qualitative shift for this culture, since extra-late flowering cultivars have spread to colder areas that had never been considered for cultivation before because of the frost risk. The Centro de Edafología y Biología Aplicada del Segura-Consejo Superior de Investigaciones Científicas (CEBAS-CSIC) Almond Breeding Program has thus achieved significant goals by releasing cultivars such as Penta, Makako and Tardona.

Tardona flowers 60 days later than traditional early cultivars such as Desmayo Largueta [2] and it is the latest flowering cultivar released to date.

Late flowering together with self-compatibility [3] are the most important traits to incorporate in new almond releases. However, the extreme delay of flowering time may have negative effects on tree productivity [4,5]. Extra- and ultra-late flowering cultivars can exhibit bud drop, floral bud abortion, irregular flowering and poor fruit set [6–8]. Low fruit set could be due to the improper development of flower buds during the previous autumn or competition with the sprouting vegetative buds [9]. Additionally, higher temperatures occurring during late flowering and early fruit development in extra-late flowering cultivars may affect stigma receptivity, shortening the effective pollination period [9].

In cold areas, it is important to breed early ripening cultivars to avoid harvesting in October, when wet and cold weather conditions can hinder fruit maturation. An extra-late flowering time could delay the ripening time, although in almond, ripening time and flowering time are apparently independent traits. Dicenta and García [10] found a low correlation coefficient between flowering and ripening time, and Sánchez-Pérez et al. [11,12] did not find a significant correlation between these traits. In fact, Desmayo Largueta has extra-early flowering but extra-late ripening. However, Tardona, ultra-late flowering, shows the same ripening time as Desmayo Largueta [2]. Finally, regarding the kernel size, some extra or ultra-late flowering cultivars with early ripening times have been found to produce small kernels. Limited kernel growth could be related to the short fruit development period or due to endocarp hardening probably promoted by high temperatures [9].

To deepen our understanding of the genetic basis of endodormancy release and flowering time in almond, the objective of this study was to modulate the flowering time by controlled temperature in order to determine the effect of flowering time on fruit set, fruit characteristics and ripening time in almond cultivars with different natural flowering times. For this purpose, almond cultivars were grown in large containers and subjected to different temperatures in controlled chambers during the endodormancy and ecodormancy periods.

## 2. Materials and Methods

### 2.1. Plant Material

The plant material assayed included the traditional Spanish extra-early flowering and self-incompatible almond cultivar Desmayo Largueta and the ultra-late flowering and self-compatible cultivar Tardona, a new release from the CEBAS-CSIC breeding program [2]. The almond cultivars Desmayo Largueta and Tardona were grafted onto GF677 rootstock clones and established in 40 L containers. The containers were placed outdoors in the Tres Caminos experimental field in Santomera (Murcia, southeast Spain) and were drip irrigated until they come in bearing in 2015, when the assays were initiated. The trees were pruned when necessary to limit the maximum height to 1.5 m. The assay was carried out for three seasons: Season 1 (2015–2016), Season 2 (2016–2017) and Season 3 (2017–2018).

### 2.2. Temperature Control for Chill and Heat Accumulation

The plants entered a dormant state in the field, before the temperature fell, in early autumn; the containers were then taken to a temperature-controlled chamber (the "Autumn" chamber) to avoid chill accumulation.

Later, two containers of each cultivar were moved weekly for 10 weeks (T1–T10) to another chamber (the "Winter" chamber) for chill accumulation. Once the cold treatment was finished, the plants (T1–T10) were weekly taken to a greenhouse (the "Spring" chamber) for heat accumulation and flowering. The cold treatments for endodormancy release and the greenhouse conditions for flowering were programmed to provoke flowering in T1 to T10 over a time period ranging from the natural flowering time of Desmayo Largueta (February 1) to that of Tardona (April 1) in our experimental field in Santomera. The experimental conditions for chill and heat accumulation were

adjusted in each season based on the results obtained in the former season. Finally, all trees were transferred to a shade shelter for vegetative growth and fruit development (Figure 1).

For each cultivar, 10 treatments (T1–T10) with two replicates (two containers) were carried out; the only exception was Desmayo Largueta T7 for which only one replicate was available in Season 3. The cold treatments for endodormancy release and the greenhouse conditions for flowering were programmed to provoke flowering in T1 to T10 over a time period ranging from the natural flowering time of Desmayo Largueta (February 1) to that of Tardona (April 1) in our experimental field in Santomera. The experimental conditions for chill and heat accumulation were adjusted in each season based on the results obtained in the former season (Figure 1).

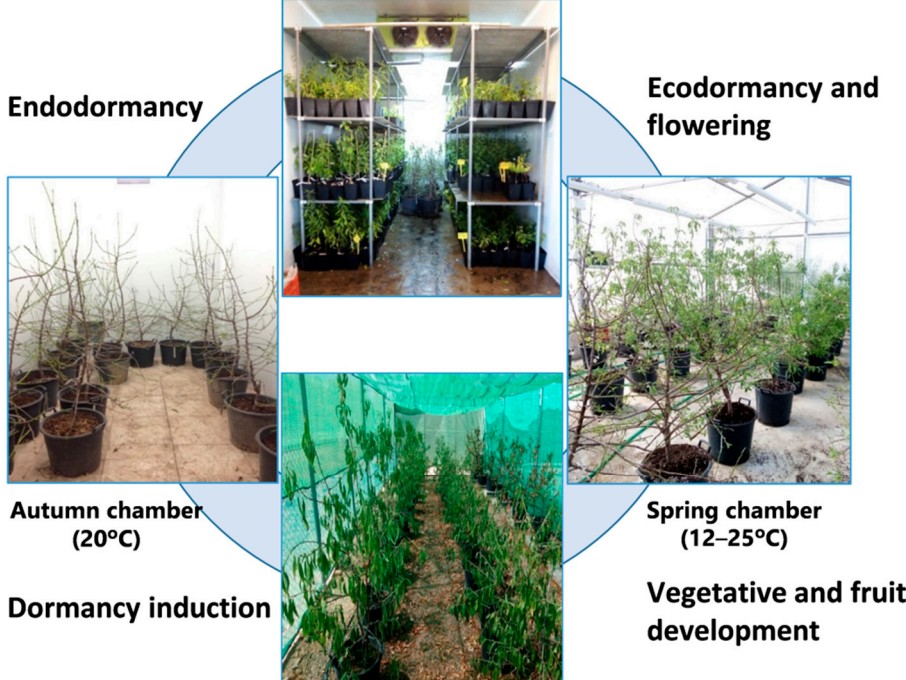

**Figure 1.** The experimental design for flowering time modulation by temperature control in different chambers.

### 2.3. Autumn Chamber

The temperature in the Autumn chamber was increased from 12–15 °C in Season 1 to 20 °C in Season 2 and Season 3. The date the plants were placed in the Autumn chamber was moved forward in each progressive season of study (November 17 in Season 1, October 14 in Season 2 and October 3 in Season 3) to allow for longer cold treatments and to prevent chill accumulation (Figure 2).

### 2.4. Chill Accumulation in the Winter Chamber

The Desmayo Largueta and Tardona treatments were placed in the Winter chamber at 7 °C on the dates indicated in Figure 2. The cold treatment necessary to fulfill the chilling requirements of each cultivar was estimated based on data obtained in the experimental field in Santomera (309 chill units (CU) for Desmayo Largueta [13] and about 1000 for Tardona). The treatments differed in terms of the moment of application and therefore in the date of endodormancy release as well (Figure 2).

The chill accumulation was estimated as chill units by applying the Richardson model [14]. The length of the cold treatments was increased from one year to another to ensure that both cultivars satisfied their chilling requirements during the trials. During the first season, 336 CU (two weeks) were applied to Desmayo Largueta and 1344 CU (8 weeks) to Tardona. During the second and third seasons,

however, the cold treatment was increased to 672 and 840 CU, respectively, for Desmayo Largueta (3 and 5 weeks) and to 1512 and 2016 CU, respectively, for Tardona (9 and 12 weeks) (Figure 2).

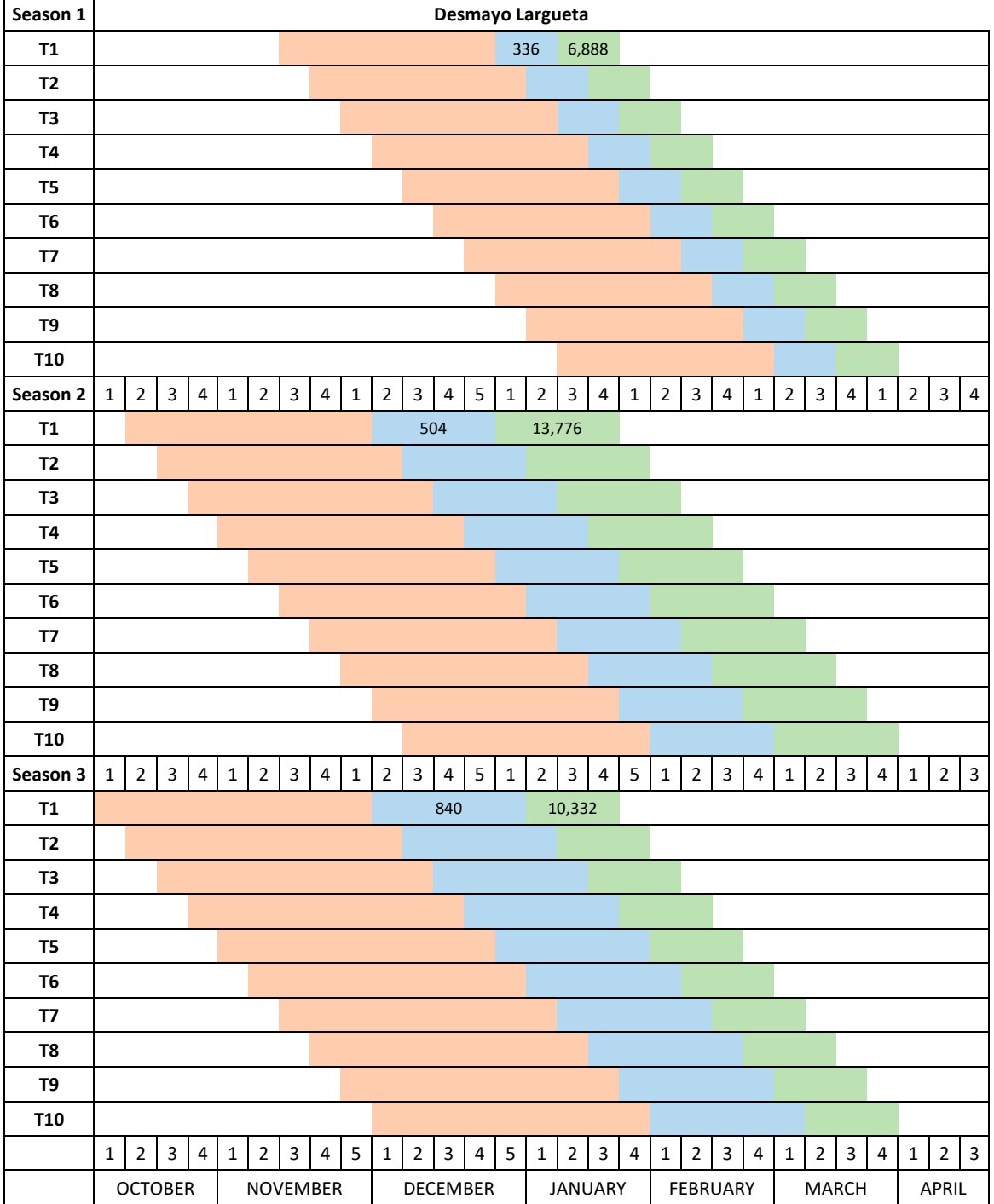

**Figure 2.** *Cont.*

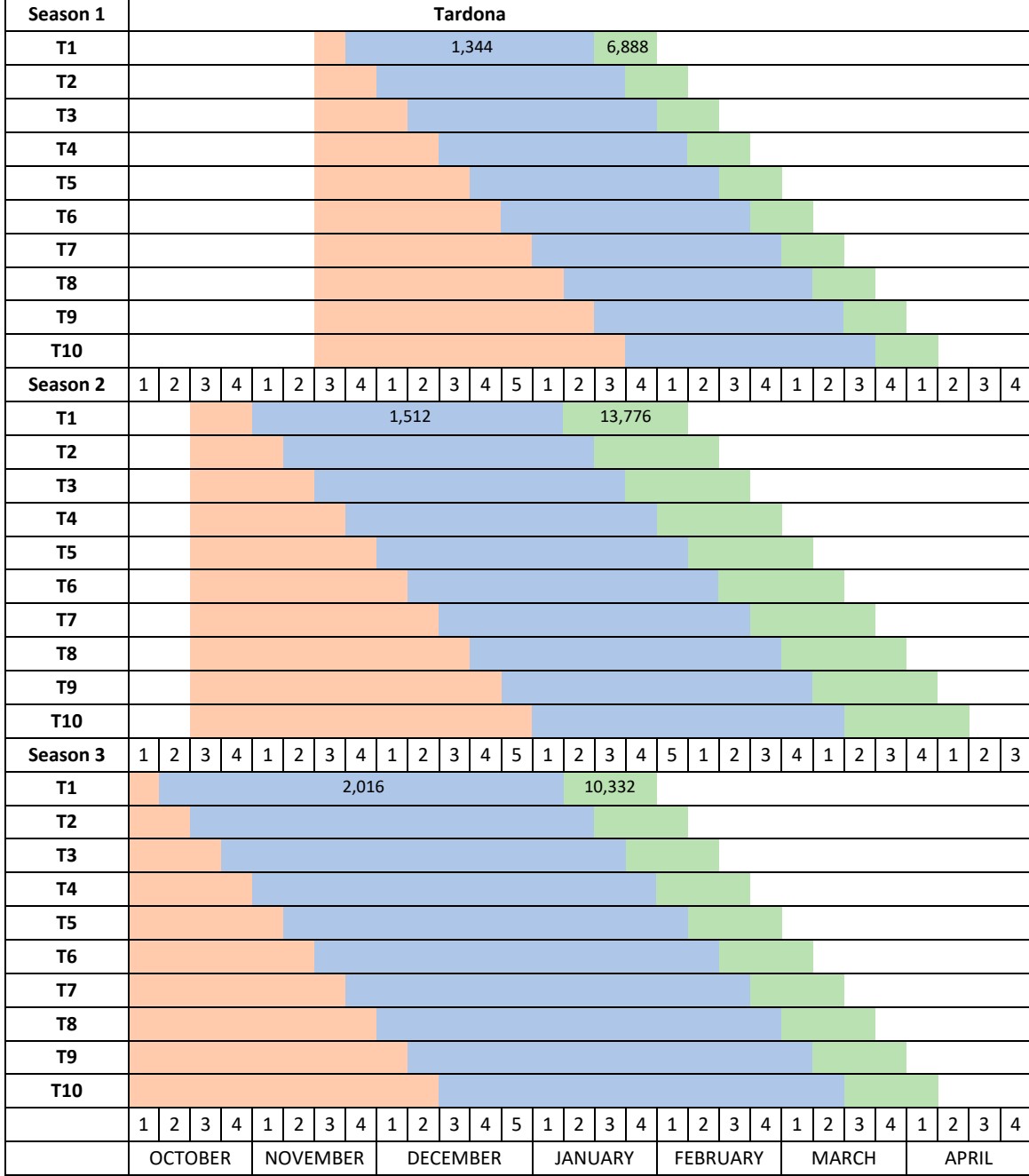

**Figure 2.** Experimental design of the temperature treatments (T1–T10) in the three controlled chambers during the three seasons studied. Autumn chamber (brown bars); Winter chamber (blue bars, with chill units accumulated); Spring chamber (green bars with growing degree hours (GDH) programmed). Months and weeks (1–5) along the treatment are indicated.

*2.5. Heat Accumulation in the Spring Chamber*

The temperature in the Spring chamber was set to 12–25 °C, although fluctuations out of this range were recorded. The heat requirements for flowering were calculated as Growing Degree Hours (GDH) according to Richardson et al. [14]. The entry dates of the treatments in the Spring chamber were earlier in Season 2 (December 28) and Season 3 (January 2) than in Season 1 (January 19), as shown in Figure 2. The estimated heat requirements for flowering increased in Seasons 2 and 3 by 6888 and 3444 GDH, respectively, with respect to Season 1 (Figure 2). The full flowering time was registered for each tree when 50% of the flowers had opened.

*2.6. Pollination and Fruit Evaluation*

In Season 1, the trees were not pollinated. In season 2 and 3, flowers from the Desmayo Largueta and Tardona cultivars were pollinated with pollen previously obtained by manual extraction from flower buds from the Achaak almond cultivar in the D state, just before opening de flower. The flowers were not emasculated in order to avoid flower damage and to promote fruit set. Once all the trees had been pollinated (a week after the last pollination), containers were carried to a shade shelter for tree and fruit development (Figure 1). The fruit set was estimated as the percentage of pollinated flowers that became fruits. The ripening time was registered when the mesocarp opened, and the mature fruits were harvested for evaluation. The fruit (in shell) and kernel weights were estimated, and the percentage of kernel (kernel weight/fruit weight) was calculated.

*2.7. Statistical Analyses*

For fruit set and fruit weight data, the means ± standard deviation (SD) were calculated for all replicates. For the fruit weight, analysis of variance was performed to test significant differences between treatments using the Kruskal–Wallis test within the agricolae package (1.3-1) (Lima, Peru) [15] in R computer language (https://cran.r-project.org/web/packages/agricolae/agricolae.pdf).

## 3. Results

*3.1. Flowering Time Modulation*

Season 1

In Season 1, Desmayo Largueta progressed from endodormancy to ecodormancy and flowered inside the Autumn chamber, so for this season, the Desmayo Largueta data were not valid. The Tardona treatments flowered gradually according to the date of entry in the Spring chamber, between March 1 and April 12. Delay mainly occurred in the first treatments, T1–T3, which flowered within the same week. These dates were later than the programmed dates (February 1 to April 1) (Figure 2).

The heat requirements for flowering ranged between 9170 and 15,037 GDH, with fluctuations of around 10,000 GDH and an average value of 11,328 GDH, with a certain downward trend from T1 to T10. Moreover, a clear decreasing trend was observed in the ecodormancy period in the Spring chamber from T1 (41 days) to T10 (21 days) (Figure 3).

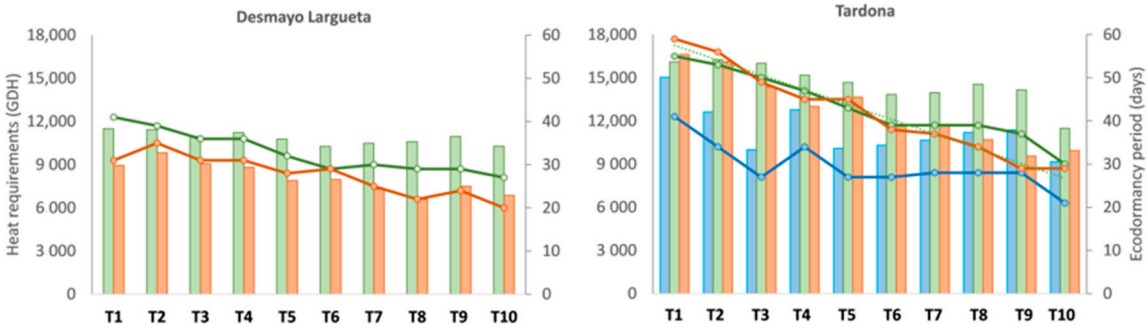

**Figure 3.** Heat requirements (bars) and the ecodormancy period (lines) during Season 1 (blue), Season 2 (green) and Season 3 (orange). Desmayo Largueta data of Season 1 were missing.

Season 2

In Season 2, the temperature in the Autumn chamber (20 °C) prevented Desmayo Largueta trees from flowering inside the chamber. The Desmayo Largueta treatments flowered between February 7 and March 25, close to the dates expected (Figure 4).

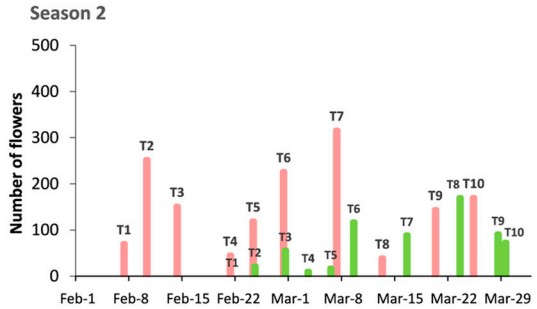
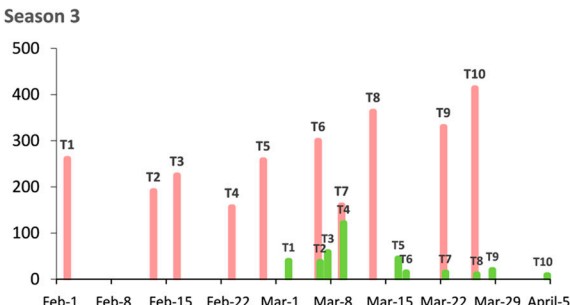

**Figure 4.** Flowering time and number of flowers pollinated of the Desmayo Largueta (pink) and Tardona (green) for T1–T10 treatments during Seasons 2 and 3.

The ecodormancy period decreased gradually from 41 to 27 days between the T1 and T10 treatments. Moreover, the GDH accumulated for flowering were highly stable, at around 11,000 GDH, with an average of 10,851 GDH (Figure 3). This means that the lower temperatures during the ecodormancy period in T1 are related to a longer ecodormancy period for flowering. The GDH accumulated were lower than the expected 13,776 (Figure 2).

In the case of the Tardona treatments, flowering took place between February 21 and March 28 (Figure 3) with an important delay in T1 and T2 with respect to the expected flowering time (February 1 to February 7) (Figure 2). The ecodormancy period also decreased from 55 to 30 days, and the GDH ranged from 16,100 to 11,488 GDH, higher than the GDH accumulated during Season 1 (Figure 3). The average was 14,623 GDH, which was higher than both expected values and the values obtained for Desmayo Largueta (Figure 3).

Season 3

In Season 3, Desmayo Largueta flowered between February 2 and March 26 with a GDH ranging from 8949 in T1 to 6865 in T10 and an ecodormancy period of between 31 and 20 days (Figure 3).

Tardona flowered between March 2 and April 4, with an important delay in the first treatments, similar to that which occurred in Seasons 1 and 2. The heat requirements gradually decreased from 16,626 to 9940 GDH, and the ecodormancy period decreased from 59 to 29 days (Figure 3).

*3.2. Fruit Set*

Generally, the fruit set ratio was lower than values usually obtained when trees are pollinated in the field, in spite of the high number of flowers pollinated (Figure 3).

In Season 2, Desmayo Largueta trees rendered 98 fruits from 1532 pollinated flowers (6%). Moreover, the fruit set tended to decrease along treatments. In the case of Tardona, 116 fruits were obtained from 645 pollinated flowers (15%).

In Season 3, the fruit set increased to 19% in the case of Desmayo Largueta (469 fruits from 2658 flowers), whereas the Tardona fruit set decreased to 7% (32 fruits from 374 flowers). The number of flowers was generally variety-dependent rather than treatment-dependent, with high variations between replicates (Figure 5). Poor flowering was observed in the ultra-late flowering cultivar Tardona during all seasons of the study compared to Desmayo Largueta (Figures 3 and 5). Anomalous flower buds and bud abscission due to the sprouting of vegetative buds were observed (Figure 6A–D). In addition, multipistil formation was commonly found in Tardona flowers under experimental conditions, a phenomenon that is not observed in the field (Figure 6E–F).

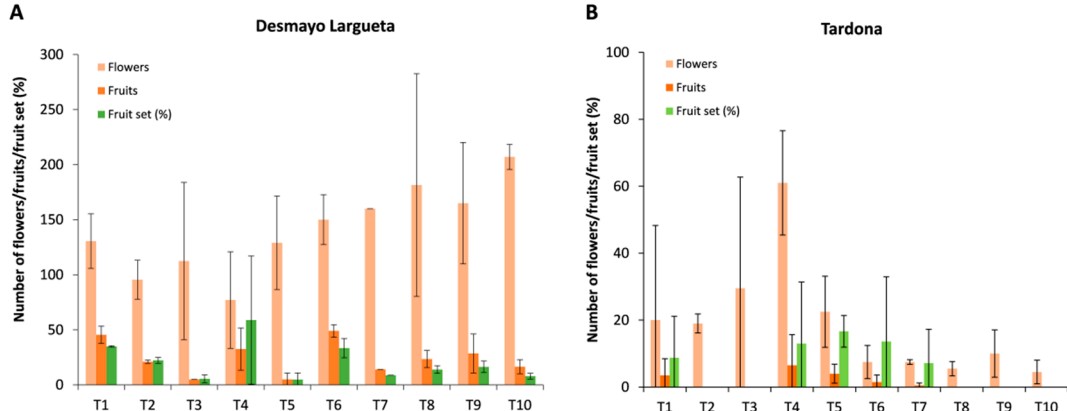

**Figure 5.** Mean number of flowers pollinated (light orange), mean number of fruits obtained (dark orange) and mean fruit set (green, %) ± SD per treatment (T) in Season 3. (**A**) Desmayo Largueta cultivar; (**B**) Tardona cultivar. Notice the different scale of A and B figures.

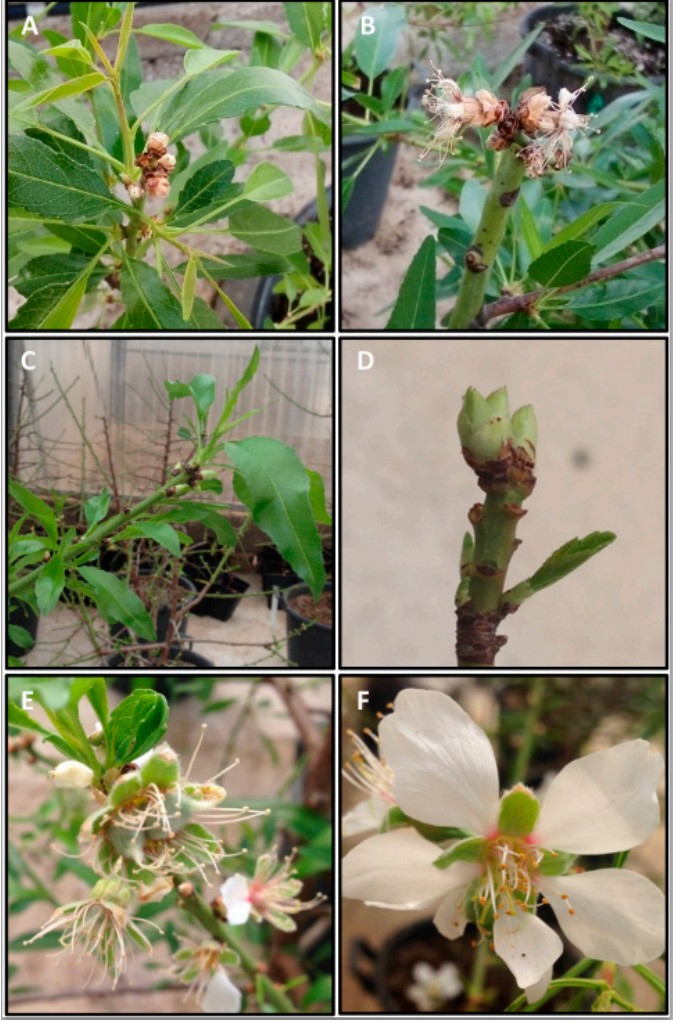

**Figure 6.** Tardona flower bud failure. Dried flower buds in different phenological states (**A,B**) and anomalous flower buds surrounded by leaves (**C,D**). Double (**E**) and multipistil (**F**) formation observed in Tardona flowers.

### 3.3. Ripening Time

In Season 2, fruits were obtained from four Desmayo Largueta treatments. T1 showed a much earlier ripening time (July 13) than that observed in the experimental field (the first week of September). A progressive delay in ripening time was observed between T2 (July 21) and T7 (August 10) (data not shown). The fruit development period, around 155 days, was similar in all treatments. In Season 3, fruits were obtained from all Desmayo Largueta treatments. Our results confirmed the advancement of the ripening time (from July 10) respect to that observed in the field, the progressive delay of ripening time according to the delay of flowering time along treatments; and the stability of the ripening period (160 days) (Figure 7).

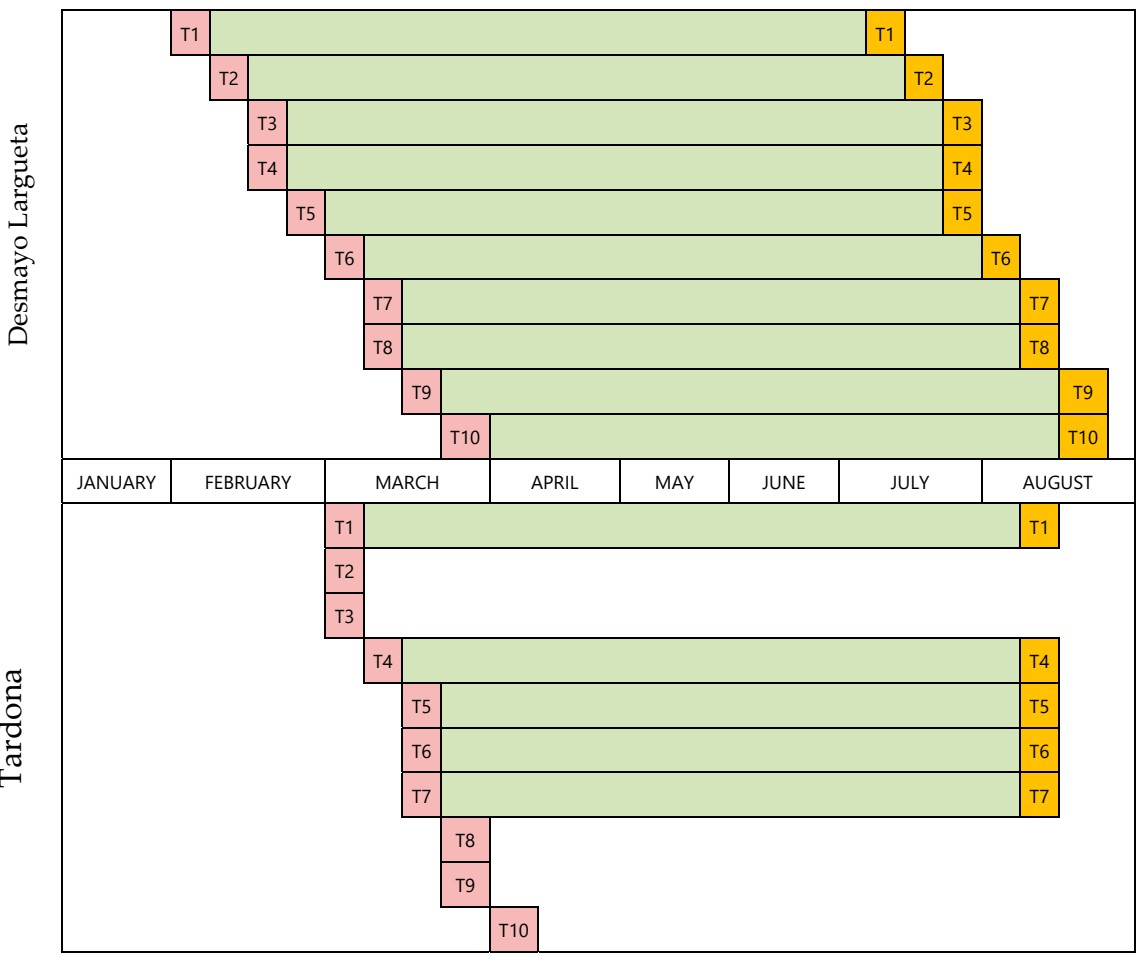

**Figure 7.** Flowering time (pink boxes) and ripening time (orange boxes) of almond treatments (T1–T10) in Season 3. The fruit development period is represented by a green bar.

Tardona fruits were obtained from T3 and T5 in Season 2, and the ripening time for both was August 10, similar to the ripening time observed in the field (August 17) (data not shown). In Season 3, Tardona fruits were obtained from five treatments and the ripening time was on August 7. The fruit development period was between 156 and 163 days in Season 2 and between 138 and 158 days in Season 3 (Figure 7).

### 3.4. Fruit Weight, Kernel Weight and Kernel/Fruit ratio

Generally, the fruits and kernels obtained from treatments were smaller than those obtained from trees grown in the field. In Season 3, the average fruit and kernel weights were 2.83 and 0.80 g, respectively, for Desmayo Largueta and 2.45 and 0.63 g, respectively, for Tardona. The values obtained

in Season 2 were slightly lower (data not shown). The experimental conditions limited the size of the fruits obtained, probably due to the high temperatures in the Spring chamber during the earliest phases of fruit development and to the culture conditions in pots.

Looking at the fruit weight results obtained in Season 3, we can see a slight increasing trend along treatments in both cultivars (Figure 8), with the exception of T7 in Tardona. This trend was not observed in kernel weight, however, which showed no significant variation along treatments. As a result, the kernel weight/fruit weight ratio showed a slight decreasing pattern from the first to the last treatment in both cultivars assayed (Figure 8).

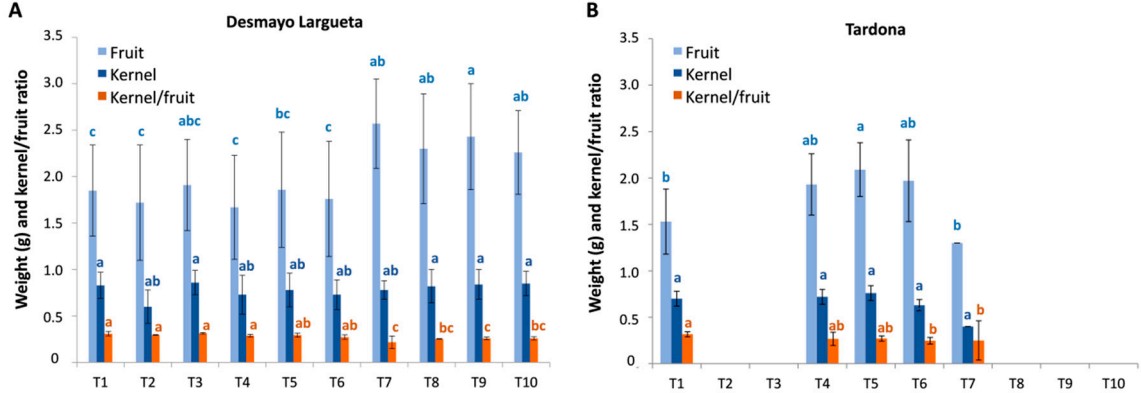

**Figure 8.** Mean weight and SD of fruits (light blue), kernel (dark blue) and kernel/fruit ratio (orange) per treatment (T) in Season 3. (**A**) Desmayo Largueta cultivar; (**B**) Tardona cultivar. Different letters indicate statistically significant differences based on the Kruskal–Wallis test ($p < 0.05$).

## 4. Discussion

The temperature treatments in controlled chambers successfully delayed the flowering time of the extra-early flowering cultivar Desmayo Largueta, except in Season 1, during which trees flowered in the Autumn chamber, where the temperature was set to 12–15 °C. According to Richardson et al. [14], temperatures between 12 and 15 °C are low enough to permit chill accumulation. The effect of temperatures on endodormancy depends on the cultivar [16], and cultivars with low chilling requirements may not enter the endodormancy state under high temperatures in autumn [17] or may enter a superficial dormancy state, which is easily overcome in moderate temperatures. During Season 2 and Season 3, the temperature of the Autumn chamber was increased to 20 °C, and the trees remained dormant and flowered after the cold treatment in the Winter chamber and the ecodormancy period in the Spring chamber. In the case of the ultra-late flowering cultivar Tardona, the flowering time could be advanced but not to the first week of February as programmed (Figure 2). Other factors apart from the chilling requirement could limit flowering in Tardona, such as the photoperiod, which was not present in the Winter chamber (continuously dark) or in the Spring chamber (natural photoperiod). In *Prunus avium*, for instance, the photoperiod was found to affect the development of winter buds at 9 °C [18]. The shorter ecodormancy period of the last treatments could be also related to the increased photoperiod and temperatures along the trial. In fact, warmer temperatures during ecodormancy could advance flowering [19].

In both cultivars, the heat accumulation for flowering in the Spring chamber was higher than in the field conditions; although, the last treatments showed similar heat requirements to those calculated in the field [1,13].

Regarding flower quality, the bud failure observed in Tardona could be explained by differences in the dormancy depth of vegetative and flowering buds. Endodormancy completion is needed for the xylem vessel differentiation that allows for the rehydration of the developing bud [20]. Vegetative buds released from endodormancy could be in a dominant state and quickly grow during ecodormancy under high temperatures [21]. This could inhibit the growth of other buds by correlative inhibitions



and lead to very short ecodormancy [7]. Additionally, it is possible that low chilling requirements cultivars such as Desmayo Largueta could better tolerate high temperatures during ecodormancy [21].

Ito et al. [22] showed that when chilling was applied too early to Japanese pear (*Pyrus communis* L.) flower buds, the effect on endodormancy release was small, even if the amount of cold applied was increased. Moreover, flower buds were found to have higher chilling requirements than vegetative buds in the study [22]. Finally, the multipistil formation commonly found in Tardona flowers under our experimental conditions may be related to the high temperatures registered in the Spring chamber [23].

## 5. Conclusions

Overall, the results obtained show that the modulation of flowering time by temperature control is possible but limited in the case of the ultra-late flowering almond Tardona. In the case of the extra-early flowering cultivar Desmayo Largueta, the delay in flowering time reflected a delay in ripening time. The fruit set and kernel weight were not variable along treatments. However, fruit weight showed a slight trend to increase and kernel/fruit ratio decreased along treatments.

Desmayo Largueta has lower heat requirements than Tardona, and the heat requirements were variable depending on the treatment and season, without any clear relationship along the treatments. Since the number of days to overcome ecodormancy decreased with the treatments, other unknown factors could be involved.

Ripening time, on the other hand, was much earlier under experimental conditions than in natural conditions for both cultivars, probably due to the culture conditions. The delay in flowering time as a result of the treatments progressively shifted the maturation time in Desmayo Largueta but not in Tardona.

The fruit set was negatively affected by the experimental conditions and it was lower than in natural conditions, especially in Tardona. No relationship was observed along the treatments applied. Finally, the nut and kernel weights were smaller than those observed in natural conditions.

**Author Contributions:** Á.S.P. and F.D. participated in the design and coordination of the study. Á.S.P. carried out the controlled temperature experiments. Á.S.P., P.M.-G. and F.D. carried out the data analysis. Á.S.P., P.M.-G. and F.D. participated in the manuscript elaboration and discussion. All authors have read and agreed to the published version of the manuscript.

**Funding:** This study has been supported by the "Almond breeding" project of the Spanish Ministry of Economy and Competiveness (grant number AGL2017-85042-R) and the project "Breeding stone fruit species assisted by molecular tools" of the Fundación Séneca of the Region of Murcia (grant number 19879/GERM/15).

**Acknowledgments:** Authors thank the technical support of Teresa Cremades Rosado during the development of the experiments.

**Conflicts of Interest:** The authors declare no conflict of interest. The funding sponsors had no role in the design of the study; in the collection, analyses, or interpretation of data; in the writing of the manuscript; or in the decision to publish the results.

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
