# Peer review of "Analysis of the Modulation of Dormancy Release in Almond (Prunus dulcis) in Relation to the Flowering and Ripening Dates and Production under Controlled Temperature Conditions"

_agronomy, doi:10.3390/agronomy10020277_

Round 1

Reviewer 1 Report

This new version is improved compared to the initially submitted manuscript. The authors correctly addressed most comments and modified the paper accordingly.

Author Response

Point 1: This new version is improved compared to the initially submitted manuscript. The authors correctly addressed most comments and modified the paper accordingly.

Response 1: Thank you very much for your comments ans suggestions to improve the manuscript.

Reviewer 2 Report

the authors have addressed my comments and the paper, as far as I am concerned, is ready to be published.

Author Response

Point 1: The authors have addressed my comments and the paper, as far as I am concerned, is ready to be published.

Response 1: Thank you very much for your comments ans suggestions to improve the manuscript.

This manuscript is a resubmission of an earlier submission. The following is a list of the peer review reports and author responses from that submission.

Round 1

Reviewer 1 Report

a)This research has been carried out with thesis that have been replicated only twice and there is not stastical analysis. On the other hand two replicas it means just one degree of freedom.

b) Figures are not very clear.

c) Fruit characterization is poor.

d) English, although generally acceptable, sometime is wrong. Needs revision by a native reviewer.

e) The major point to me is that the premises of the research are poor since are very well known since years.

Author Response

Point 1: This research has been carried out with thesis that have been replicated only twice and there is not stastical analysis. On the other hand two replicas it means just one degree of freedom.

Response 1: Statistical analysis has been performed for fruit weight, kernel weight and yield data coming from a variable number of fruits from each container per treatment. No correlations were found between treatments and number of flowers, number of fruits and fruit set, because of the low number of replicates, and data distributed randomly.

Point 2: Figures are not very clear.

Response 2: Figures were changed to a higher resolution for a clear visualization.

Point 3: Fruit characterization is poor.

Response 3: This study focused on the characterization of fruit weight, as an indicator of fruit size, because of the small size observed in some late flowering almond cultivars.

Point 4: English, although generally acceptable, sometime is wrong. Needs revision by a native reviewer.

Response 4: The manuscript has been strongly revised by a native reviewer. Due to the great quantity of performed changes NO CONTROL CHANGES HAVE BEEN USED. Practically the manuscript has been rewritten.

Point 5: The major point to me is that the premises of the research are poor since are very well known since years. 

Response 5: Although it is known that ripening time is not related to flowering time in almond, we observed an advance of ripening time (that implied a shortening of the production cycle) under our experimental conditions, and a shift of ripening time in response to the delay of flowering time along treatments, in case of the early flowering almond cultivar ‘Desmayo Largueta’.

Reviewer 2 Report

Many parts should be dramatically improved. the overall presentation of the manuscript is faulty. Some sentences in the abstract are (lines 19-21) are unclear, others are commonplace (lines 25-28). It is well known that ripening time is not correlated with flowering time in many fruit species, like grape, olive, apple,  yet the authors hypothesize that may be the case in almond (p. 2, lines 50-52). There are methodological problems. The authos used two replicates per treatment, there is no statistical analysis of data, the plant material is  poorly described (were cultivars self-compatible? how high were the plants each year? were they trained to a single stem?), fruit quality determinations were basic and could have been more extensive, legends and captions in figures are not easy to read or understand (there are mistakes too, springer instead of spring, are pots in winter chamber 40 L pots? Natural conditions? Plants were kept in a lathhouse!). The English should be definitely improved.

Author Response

Point 1: Many parts should be dramatically improved. the overall presentation of the manuscript is faulty.

Response 1: The manuscript structure has been corrected and mainly abstract, results, and discussion sections rewritten. Due to the great quantity of performed changes NO CONTROL CHANGES HAVE BEEN USED. Practically the manuscript has been rewritten.

Point 2: Some sentences in the abstract are (lines 19-21) are unclear, others are commonplace (lines 25-28)..

Response 2: Abstract lines 19-21 have been clarified. Abstract lines 25-28 were changed to provide interesting suggestions derived from results.

Point 3: It is well known that ripening time is not correlated with flowering time in many fruit species, like grape, olive, apple, yet the authors hypothesize that may be the case in almond (p. 2, lines 50-52)..

Response 3: Although it is known that ripening time is not related to flowering time in almond, we observed an advance of ripening time (that implied a shortening of the production cycle) under our experimental conditions, and a shift of ripening time in response to the delay of flowering time along treatments, in case of the early flowering almond cultivar ‘Desmayo Largueta’.

Point 4: There are methodological problems. The authors used two replicates per treatment, there is no statistical analysis of data.

Response 4: Statistical analysis has been performed for fruit weight, kernel weight and yield data coming from a variable number of fruits from each container per treatment. No correlations were found between treatments and number of flowers, number of fruits and fruit set, because of the low number of replicates, and data distributed randomly.

Point 5: The plant material is poorly described (were cultivars self-compatible? how high were the plants each year? were they trained to a single stem?). 

Response 5: Plant material section has been completed with self-compatibility information and pruning performed on trees in containers.

Point 6: Fruit quality determinations were basic and could have been more extensive. 

Response 6: This study focused on the characterization of fruit weight, as an indicator of fruit size, because of the small size observed in some late flowering almond cultivars.

Point 7: legends and captions in figures are not easy to read or understand (there are mistakes too, springer instead of spring, are pots in winter chamber 40 L pots? Natural conditions? Plants were kept in a lathhouse. 

Response 7: Legends were improved and mistake was corrected. In the picture of the Winter chamber 40L can be observed together with other smaller pots. Natural conditions referred to field conditions during fruit development because during this phase trees were kept in a shade shelter in the field.

Point 8: The English should be definitely improved. 

Response 8: The manuscript has been revised by a native reviewer.